# Delays in the presentation and diagnosis of women with breast cancer in Yogyakarta, Indonesia: A retrospective observational study

**Susanna Hilda Hutajulu**[1]*, **Yayi Suryo Prabandari**[2,3], **Bagas Suryo Bintoro**[2,3], **Juan Adrian Wiranata**[4], **Mentari Widiastuti**[3], **Norma Dewi Suryani**[5], **Rorenz Geraldi Saptari**[6], **Kartika Widayati Taroeno-Hariadi**[1], **Johan Kurnianda**[1], **Ibnu Purwanto**[1], **Mardiah Suci Hardianti**[1], **Matthew John Allsop**[7]

1 Division of Hematology and Medical Oncology, Department of Internal Medicine, Faculty of Medicine, Public Health and Nursing, Universitas Gadjah Mada/Dr Sardjito General Hospital, Yogyakarta, Indonesia, 2 Department of Health Behaviour, Environment, and Social Medicine, Faculty of Medicine, Public Health and Nursing, Universitas Gadjah Mada, Yogyakarta, Indonesia, 3 Center of Health Behaviour and Promotion, Faculty of Medicine, Public Health and Nursing, Universitas Gadjah Mada, Yogyakarta, Indonesia, 4 Medical Internship Program, Academic Hospital, Universitas Gadjah Mada, Yogyakarta, Indonesia, 5 Division of Hematology and Medical Oncology, Department of Internal Medicine, Dr Sardjito Hospital, Yogyakarta, Indonesia, 6 Medicine Study Program, Faculty of Medicine, Public Health and Nursing, Universitas Gadjah Mada, Yogyakarta, Indonesia, 7 Leeds Institute of Health Sciences, School of Medicine, Faculty of Medicine and Health, University of Leeds, Leeds, United Kingdom

* susanna.hutajulu@ugm.ac.id

**Data Availability Statement:** We have revised the dataset to now include deidentified data used to generate Fig 1 and Tables 1–6. We have provided

## Abstract

### Purpose

To investigate factors associated with delays in presentation and diagnosis of women with confirmed breast cancer (BC).

### Methods

A cross-sectional study nested in an ongoing prospective cohort study of breast cancer patients at Dr Sardjito Hospital, Yogyakarta, Indonesia, was employed. Participants (n = 150) from the main study were recruited, with secondary information on demographic, clinical, and tumor variables collected from the study database. A questionnaire was used to gather data on other socioeconomic variables, herbal consumption, number of healthcare visits, knowledge-attitude-practice of BC, and open-ended questions relating to initial presentation. Presentation delay (time between initial symptom and first consultation) was defined as ≥3 months. Diagnosis delay was defined as ≥1 month between presentation and diagnosis confirmation. Impact on disease stage and determinants of both delays were examined. A Kruskal-Wallis test was used to assess the length and distribution of delays by disease stage. A multivariable logistic regression analysis was conducted to explore the association between delays, cancer stage and factors.

derived values for date information to avoid identification of participants. Sharing of the full de-identified dataset is not possible due to restrictions imposed by the ethics committee as most of these contain patient data, albeit de-identified, and it may be possible to determine the identity of participants given the extent of sociodemographic and clinical data available for each participant. Should there be a request for data, this can be sent to the corresponding author (email: susanna.hutajulu@ugm.ac.id). Future researchers can contact the institutional ethics committee (email: mhrec_fmugm@ugm.ac.id) at Universitas Gadjah Mada, Indonesia with data access queries as well.

**Funding:** SHH received funding from Kementrian Riset, Teknologi dan Pendidikan Tinggi Republik Indonesia (ID) (2018) and Universitas Gadjah Mada (2021). MJA received funding from University of Leeds (2019). The funders had no role in study design, data collection and analysis, decision to publish, or preparation of the manuscript.

**Competing interests:** The authors have declared that no competing interests exist.

## Results

Sixty-five (43.3%) patients had a ≥3-month presentation delay and 97 (64.7%) had a diagnosis confirmation by ≥1 month. Both presentation and diagnosis delays increased the risk of being diagnosed with cancer stage III-IV (odds ratio/OR 2.21, 95% CI 0.97–5.01, p = 0.059 and OR 3.03, 95% CI 1.28–7.19, p = 0.012). Visit to providers ≤3 times was significantly attributed to a reduced diagnosis delay (OR 0.15, 95% CI 0.06–0.37, p <0.001), while having a family history of cancer was significantly associated with increased diagnosis delay (OR 2.28, 95% CI 1.03–5.04, p = 0.042). The most frequent reasons for delaying presentation were lack of awareness of the cause of symptoms (41.5%), low perceived severity (27.7%) and fear of surgery intervention (26.2%).

## Conclusions

Almost half of BC patients in our setting had a delay in presentation and 64.7% experienced a delay in diagnosis. These delays increased the likelihood of presentation with a more advanced stage of disease. Future research is required in Indonesia to explore the feasibility of evidence-based approaches to reducing delays at both levels, including educational interventions to increase awareness of BC symptoms and reducing existing complex and convoluted referral pathways for patients suspected of having cancer.

## Introduction

Breast cancer has the highest cancer incidence in Indonesia and the second highest cause of cancer mortality in females. In 2018, the annual estimated incidence and mortality per 100,000 individuals were 44 and 15.3 per year respectively [1]. In the region of Yogyakarta, Indonesia, breast cancer patients are generally diagnosed at stages III and IV [2, 3]. The 5-year overall survival rate is generally unfavourable. It is 48–50% for the whole disease spectrum [2, 4] and 12% for those with advanced diseases [4].

Delays in breast cancer presentation and diagnosis are likely to be key factors in advanced-stage presentation [5]. There are disparities in the length of delays between countries. Reports from high-income countries report median times of 14–60 days [6–8], with a presentation delay of >3 months occurring in 17–35% of patients [6, 7, 9–11]. Reports from low- and middle-income countries report a longer length of delays. A study conducted in the neighbouring country of Malaysia found that the median time to consultation and diagnosis was 2 months and 5.5 months, respectively [12]. In Rwanda, a median time of 5 months for both presentation and diagnosis delays were observed [13]. In Indonesia, previous studies have observed >3-month presentation delay in 36.2% of cases and >1-month diagnosis delay in 25% of cases [14]. Furthermore, a 7-month median time of presentation delay and a 6-month median time of the delay to commencing treatment, relating to the time taken from diagnosis to initial treatment, have been observed [15].

Many sociodemographic factors, clinical factors, and patients' experiences have been reported as influencing presentation delay. Age, residence, distance to a medical facility, marital status, education level, occupation, insurance, health facility visits, visiting traditional medicine practitioners, knowledge of breast cancer, breast self-examination, initial symptom, family history of breast cancer, and comorbidities are factors that have been associated with delays in both presentation and subsequent diagnosis [5, 6, 10, 13, 16–19]. Reasons for delays

include lighter symptoms, fear of informing other people, negative attitudes toward medical health professionals and fear of treatment [13, 20]. In a local publication, initially consulting with non-medical practitioners for breast-related complaints and consuming non-medical treatments have also been associated with diagnosis and treatment delays [14]. Observed by qualitative studies, a lack of awareness and knowledge of cancer, cancer beliefs, treatment beliefs, financial problems, emotional burden, paternalistic style of communication, and unmet information needs are related to psychosocial and cultural reasons for patient delay [15, 21, 22].

To date, research focusing on delays in presentation and diagnosis for patients with breast cancer [15, 22, 23] in Indonesia have included qualitative studies [15, 23, 24], including a study undertaken in Yogyakarta, the region in which this study is focused [21]. There is very limited research quantifying factors affecting delays in Indonesia [14, 15], with none exploring factors in Yogyakarta. The Special Region of Yogyakarta, with a current population of 3,882,288 [24], has the highest frequency of cancer in the country [25], with breast cancer being the commonest malignancy [26]. The objective of this study is to quantitatively investigate the factors associated with presentation and diagnosis delays, the relationship of delays to stage at presentation and reasons for patient delay within local breast cancer cases.

## Method

### Study subject

This cross-sectional nested study recruited 150 Indonesian female breast cancer patients registered in a prospective ongoing main study. The main study aims to analyse the risk of side effects from chemotherapy and determine their effect on the survival and quality of life in 250 breast cancer patients. The registered participants are patients visiting and receiving their first chemotherapy treatment in the Hematology and Medical Oncology Division, "Tulip"/Integrated Cancer Clinic, Dr. Sardjito General Hospital, Yogyakarta, Indonesia, from 2018–2022. Women aged ≥18 years with histopathologically confirmed breast cancer without terminal condition and severe congestive heart failure have been recruited. Cases received chemotherapy as neoadjuvant treatment (before surgery), adjuvant treatment (after surgery), or palliative treatment with or without surgery. Study subjects were contacted and approached to participate in the cross-sectional observation study with consent before recruitment.

### Method of data collection

From the main study database, we collected secondary information on demographic variables (age, education, and residence), comorbidity, family history of cancer, date of the first symptom, date of first medical visit, nutritional status (determined by body mass index/BMI), first symptom of breast cancer, and cancer staging upon diagnosis.

To acquire determinants that were unavailable initially in the study database, we developed a questionnaire with 27 questions in the initiation phase. Questions that were listed in the questionnaire were adapted from multiple sources, including the Indonesian Family Life Survey Wave 5 [27] for living arrangement, socioeconomic status and accessibility to the first medical facility visit and Breast Module of the Cancer Awareness Measure (Breast-CAM) for knowledge of breast cancer risk factors and habit of breast self-exam [28]. We also adapted variables from existing studies to develop questionnaire items [13, 17, 29, 30]. Questionnaire development started with the identification of variables that will be measured and the items from existing questions that related to the selected variables. Items were further selected before the translation process. Three independent translators performed forward-backward translation to produce a questionnaire with semantic equivalence to the original versions in English.

Assessment of validity included face validity and questionnaire finalization (see S1 Fig). Face validation was conducted through pilot testing the questionnaire with six lay members of the public, convenience sampled through contacts known to the research team. Three women were primary care facility patients who came for a routine appointment, two women were primary care facility administrative officers, and one woman was a household assistant for a research team member. We conducted a discussion with the participants and gained helpful suggestions for determining which questions required refinement. Through this process, from an initial set of 27 questions, one question about the type of facilities visited to check their symptoms was removed due to redundancy. We also added nine questions to accommodate issues that had not been covered in the initial set of questions. These questions were about patients' address based on their ID card, whether patients still lived at the same address, date of initial symptom, date of first medical contact, distance from their place of residence to the site of first medical contact, and four questions about herbal medicine consumption (the type, dose, and frequency of consumption). While the date of the first breast cancer symptom was already provided in the database, we asked it again in the questionnaire to confirm the information. Five questions were re-worded to improve readability and clarity, and open-ended questions were included at the end of the questionnaire to elicit participants' reasons for presentation delay. The process of face validity resulted in a total set of 35 questions.

From September 2020 to February 2021, a face to face interview was conducted with the participants. Trained researchers performed the interview at the ambulatory clinic for 77 participants and, due to the pandemic situation, through phone calls for 74 participants. The questionnaire was only administered once patient informed consent had been obtained. The joint ethics committee from the Faculty of Medicine, Public Health and Nursing, Universitas Gadjah Mada/Dr Sardjito General Hospital, Yogyakarta has approved the main ongoing prospective study (reference number: KE/FK/0417/EC/2018) and provided specific ethical approval for the study reported in this manuscript (amendment from the ongoing study with reference number KE/FK/0444/EC/2020).

## Delays definition

Presentation delay was defined as the number of months between breast symptom onset and the patient's first presentation to a medical professional (doctor, nurse, or midwife). Diagnosis delay was used to define the number of months between the first visit to a medical professional and the date of the first pathology report confirming a diagnosis of breast cancer. When respondents could not remember the date of when the symptoms first appeared or the date when they first visited a health facility, we asked for a time span in months. After providing the range of months, we sought to direct the patient to remember the distance between those dates from important dates/events such as family/respondent's birthdays, or religious holidays, to further narrow the range and improve the recall closer to the exact date. When provided with a single month, we tried to ask about the exact day or its distance with other important events in the month, to further narrow the date into a single exact day. We also set breast surgery/procedure dates that were recorded in clinical notes as a benchmark, because it was considered by most patients as an important event. Participants were asked the date of first symptom and first medical visit, based on surgery date as a benchmark. When respondents were unable to provide a date for when their symptoms began or the first provider visit, they were asked to provide a month or month range and year. If they provided a month, the date was estimated as the 15th of that month; if they provided a month range, the estimated date was the midpoint between the 15th of those months. If patients were only able to provide a year, the estimated date was June 30th of that year [13].

An agreement was decided when there was conflicting information of the date of first symptoms in the existing database and that was collected from the interview. In such a case, we relied on the self-report data in the database. We used a delay of ≥3 months to define presentation delay based on substantial evidence that such delays are associated with lower survival. We used ≥1 month to define diagnosis delay because one month is indicated as an adequate time for the physician to take appropriate action and shorter delays are not clinically significant [20].

## Key independent variables

The key independent variables included various parameters that were obtained from the existing study database and the interview. Sociodemographic information included age at diagnosis, residence (urban or rural, referred to the regional statistical bureau), distance to the first health facility visited (<3 km or ≥3 km), living arrangement (living alone, with spouse only, with other than a spouse, or with spouse and others), educational attainment (<junior high school or ≥junior high school, referred to the 9-years compulsory education in Indonesia), monthly household income (<3,000,000 or ≥3,000,000 Indonesian Rupiahs/IDR, referred to a happiness index by the Indonesian Ministry of National Development Planning in 2014), type of medical professional first visited (general practitioners in a public primary health care facility or private practices, a specialist doctor in a public hospital or private practices, or midwife or others), and frequency of medical visit after first consultation (<3 time or ≥3 times). We also collected information on the use of herbal medicine before the first consultation after breast problem recognition. Knowledge about breast cancer risk factors was interviewed and classified as not know any risk factor or know at least one of the risk factors that are determined elsewhere [31]. The frequency of breast self-exam was categorized as rarely or never, at least once every 6 months, at least once a month, and at least once a week. Features of the patient's experience with the breast problem were categorized as breast lump, other complaint, and breast lump and other complaint. Family history of cancer (none or presence) and the presence of comorbidities (none, 1, or ≥2) were categorized based on self-report from the existing database. Comorbidities included diabetes, hypertension, hepatitis, heart failure, or other health problems. Body mass index (BMI) was calculated from measured weight and height. We used the World Health Organization Asia-Pacific body mass index classifications, which classified BMI into underweight (<18.5), normal (18.5–22.9), overweight (23–24.9) and obese (≥25). For analysis purposes, we further stratified BMI into two groups: underweight to normal (<23) and overweight to obese (≥23). Information of performance index (determined as ECOG 0–1 or ≥2) was obtained from the database. Breast cancer was staged using the 7th edition AJCC staging system and then simplified as early (stage I-II), locally advanced (stage III) and metastatic (stage IV) disease [32]. A content analysis of the interview transcripts [33] was performed to assess the patients' reasons for a delay in presentation. We coded the interview transcripts by open coding, organising responses into a meaningful set of categories that covered all relevant data.

## Statistical analyses

Kruskal-Wallis statistical test was performed to assess the lengths of presentation and diagnosis delay among the patients and distribution of delays by stage at presentation. This analysis was chosen as the histogram of delay data were not normally distributed. We visually inspected the distribution of the data using histograms for data on both diagnosis time and presentation time which showed data for both variables were skewed to the right. A multivariable logistic regression analysis was conducted to elucidate the factors that are attributable to both

presentation and diagnosis delays. Firstly, we conducted logistic regression to explore the association of each factor to presentation and diagnosis delays. Variables with a p-value <0.25 in the univariate logistic regression plausibly related to a presentation or diagnosis delay were included in the multiple logistic regression models. Variables that correlated highly with another variable in the model was removed. A similar approach was used to explore the association between presentation and diagnosis delay with cancer stage at diagnosis. The statistical significance was based on a two-sided p-value of <0.05. Statistical analysis was done using Stata version 14/15 (Stata Corp).

## Results

### Subject characteristics

When the present study started, 214 have been registered in the main study. Three cases dropped out and 38 cases had died. From the 173 eligible cases, 16 cases did not respond to our invitation and 6 cases refused to participate. Finally, 151 subjects were interviewed (recruitment rate 87.3%) with a questionnaire response rate of 100%. One patient was excluded due to participants' inability to recall and communicate well in the interviews. Characteristics of all subjects are summarized in Table 1. The cohort was dominated by those living in the urban area (105, 70%), living within ≤3 km from the health facility they firstly visited (87, 57.6%), living with their spouse and other family members (93; 61.6%), having an education at least junior high school (88; 58.67%), possessing lower monthly income (<3,000,000 IDR) (91; 60.7%), experiencing ≤3 times of medical visit before diagnosis (91; 60.7%), visiting general practitioners as the first medical contact for breast symptoms (88; 58.7%), taking alternative medicine before presentation (106; 70.7%), having enough knowledge of breast cancer risk factor (106; 70.7%), and rarely or never performing breast self-examination (82; 54.7%). The majority of participants experienced breast lump as a first complaint (101; 67.3%), did not have a family history of cancer (93; 62%), and presented with no comorbidity (66; 44%). Upon diagnosis, the majority of patients had overweight to obese BMI (82; 54.7%), good physical performance (ECOG 0–1) (145; 96.7%), presented with stage III disease (67; 44.7%), and had the luminal B subtype (62; 41.3%). While being asked about the time when the symptoms first appeared, 13 (8.7%) provided the exact date, 2 (1.3%) provided the date range, 80 (53.3%) provided the month and year, 5 (3.3%) provided a month range and year, and 50 (33.3%) only provided a year. Regarding the date of the first visit to the health facility, 25 (16.7%) provided the exact date, 15 (10%) provided the date range, 85 (56.7%) provided the month and year, 8 (5.3%) provided a month range and year, and 17 (11.3%) only provided a year.

### The magnitude of delays and their influence on disease stage

The median time to presentation from initial symptoms experienced by participants was 2 months (61 days) (Fig 1). Eighty-five (56.7%) respondents had a consultation with a medical professional within 3 months after detecting their symptoms, while 65 (43.3%) delayed the consultation by ≥3 months. The median time to diagnosis confirmation from first consultation experienced by participants was 1 month. As many as 53 (35.3%) respondents had their breast cancer diagnosed within 1 month while 97 (64.7) participants had confirmation by ≥1 month. Overall, the median time to diagnosis from initial symptom was 7 months.

Median presentation time for patients with early, locally advanced and metastatic disease was 0, 2, and 9.5 months respectively (p <0.001). The median diagnosis time for the three groups was 0, 2, and 1.5 months respectively (p = 0.006). Median time from initial symptom to diagnosis (overall time) was 3, 10, and 25 months among patients with stage I/II, III, and IV disease (p <0.001) (Table 2).

**Table 1. Sociodemographic, clinical, and delay characteristics of study participants (n = 150) recruited from Dr. Sardjito Hospital, Yogyakarta in 2018–2021.**

| Variables | Frequency n (%) |
|---|---|
| Age (years) | |
| Mean±SD | 52.1±9.0 |
| Residence | |
| Urban | 105 (70) |
| Rural | 45 (30) |
| Distance to the first medical visit | |
| ≤3 km | 87 (58) |
| >3 km | 63 (42) |
| Living arrangement | |
| With spouse only | 19 (12.7) |
| With other than spouse | 32 (21.3) |
| With spouse and other | 93 (62) |
| Living alone | 6 (4) |
| Education status | |
| ≥junior high school | 88 (58.7) |
| <junior high school | 62 (41.3) |
| Monthly income (IDR) | |
| ≥3,000,000 | 59 (39.3) |
| <3,000,000 | 91 (60.7) |
| Frequency of medical visit before diagnosis | |
| >3 times | 59 (39.3) |
| ≤3 times | 91 (60.7) |
| Types of health care facility first visited | |
| Specialist (private or public) | 45 (30) |
| General practitioner (private or public) | 88 (58.7) |
| Midwife and other | 17 (11.3) |
| Consumption of herbal medicine before presentation | |
| No | 44 (29.3) |
| Yes | 106 (70.7) |
| Knowledge of risk factors of breast cancer | |
| Diet-related | 90 (60) |
| Air pollution | 14 (9.3) |
| Smoking or being a second-hand smoker | 28 (18.7) |
| Alcohol consumption | 4 (2.7) |
| Stress | 20 (13.3) |
| Radiation | 1 (0.7) |
| Exhaustion or sleep deprivation | 6 (4) |
| Genetic | 41 (27.3) |
| Hormonal compound from contraception use | 20 (13.3) |
| Breastfeeding | 6 (4) |
| Less exercise | 3 (2) |
| Other | 11 (7.3) |
| Number of risk factor of breast cancer known | |
| ≥1 | 106 (70.7) |
| None | 44 (29.3) |
| Habit of breast self-examination | |

(*Continued*)

**Table 1.** (Continued)

| Variables | Frequency n (%) |
|---|---|
| At least once a week | 42 (28) |
| At least once a month | 18 (12) |
| Rarely or never | 90 (60) |
| First presenting symptom | |
| Breast lump only | 101 (67.3) |
| Breast lump and other | 35 (23.3) |
| Other than breast lump | 14 (9.3) |
| Family history of cancer | |
| No | 93 (62) |
| Yes | 57 (38) |
| Types of comorbidity | |
| No comorbidity | 66 (44) |
| Hypertension | 38 (25.3) |
| Diabetes | 14 (9.3) |
| Dyslipidemia | 9 (6) |
| Other | 44 (29.3) |
| Number of comorbidities | |
| None | 66 (44) |
| 1 | 63 (42) |
| ≥2 | 21 (14) |
| BMI (WHO Asia-Pacific) | |
| <23 | 68 (45.3) |
| ≥23 | 82 (54.7) |
| ECOG performance status | |
| 0–1 | 145 (96.7) |
| ≥2 | 5 (3.3) |
| Stage | |
| I–II | 53 (35.3) |
| III | 67 (44.7) |
| IV | 30 (20) |

Abbreviations: SD = Standard Deviation; km = Kilometre; IDR = Indonesian Rupiah; BMI = Body Mass Index; ECOG = Eastern Cooperative Oncology Group.

### Factors that are attributable to both presentation and diagnosis delay

Tables 3 and 4 showed the multivariate analyses of sociodemographic, clinical, and tumor characteristics that may influence presentation and diagnosis delay. Monthly income of <3,000,000 IDR is a factor associated with an increased delay in presentation with marginal significance (odds ratio/OR 2.21, 95% confidence interval/CI 0.99–4.93, p = 0.052). Patients who visited healthcare facilities ≤3 times before diagnosis had a reduced risk of experiencing diagnosis delay (OR 0.15, 95% CI 0.06–0.37, p <0.001). Having a family history of cancer is a significant factor that is related to a higher risk of diagnosis delay (OR 2.28, 95% CI 1.03–5.04, p = 0.042).

### Effect of delay on the likelihood of worse clinical presentation

The effect of delays on presenting with a more advanced stage at baseline was displayed in Table 5. We conducted two analysis models for each type of delay to investigate the correlation

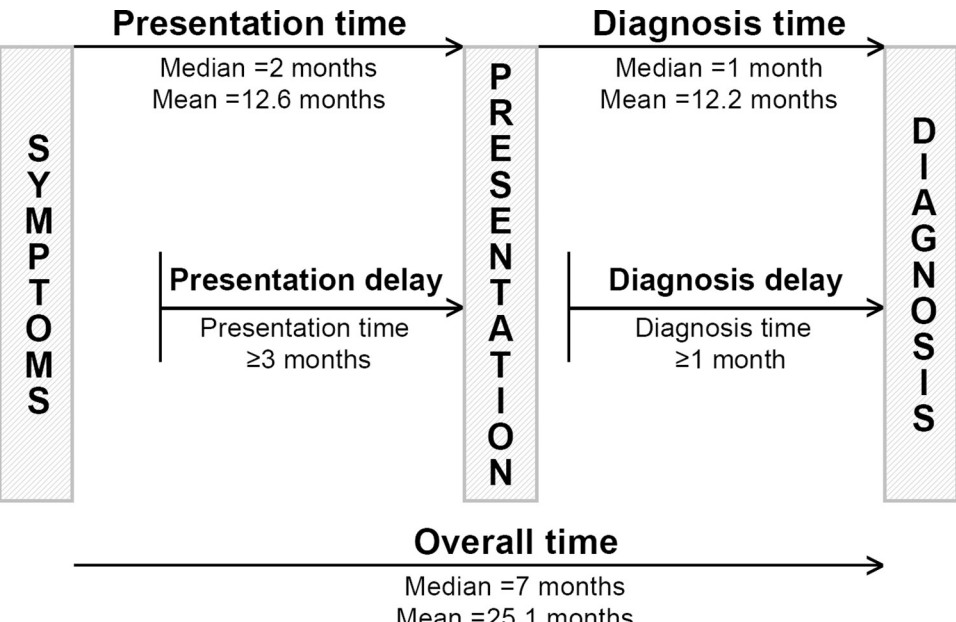

**Fig 1. Timeline sketch of breast cancer presentation and diagnosis.** Participants were observed to have a 2-month median presentation time and 1-month diagnosis time. The median time to diagnosis from initial symptom was 7 months. 43.3% of respondents delayed the consultation by ≥3 months and 64.7% had diagnosis confirmation by ≥1 month.

between delays and presenting with a more advanced stage at diagnosis. Model 1 for presentation delay is adjusted for education level, monthly income, number of risk factors of breast cancer known, habit of breast self-exam, first presenting symptom and number of comorbidities. Model 1 for diagnosis delay is adjusted for education level, monthly income, frequency of medical visit before diagnosis, number of risk factors of breast cancer known, habit of breast self-exam, first presenting symptom and number of comorbidities. Model 2 for presentation delay is an analysis of model 1 for presentation delay which also adjusted with diagnosis delay. Model 2 for diagnosis delay is an analysis of model 1 for diagnosis delay which is also adjusted with presentation delay. When focused on model 2, ≥3 months of delay in the presentation was associated with an increased risk of having a more advanced stage disease (III-IV) (OR 2.21, 95% CI 0.97–5.01, p = 0.059), although significance was not reached. Patients with a delay in diagnosis of ≥1 month were more likely to present with a more advanced stage (OR 3.03, 95% CI 1.28–7.19, p = 0.012). When extended to various presentation and diagnosis time, it is demonstrated that more participants with longer delays presented with more advanced stages (see S1 Table).

**Table 2. Median presentation, diagnosis, and overall delay for all participants (n = 150) by stage at diagnosis.**

| Duration of delay (months)[+] | Stage I-II (n = 53) | Stage III (n = 67) | Stage IV (n = 30) | Total (n = 150) | P unadjusted[*] |
|---|---|---|---|---|---|
| Presentation time | 0 (0–3) | 2 (0–10) | 9.5 (1–24) | 2 (0–9) | <0.001 |
| Diagnosis time | 0 (0–2) | 2 (1–6) | 1.5 (0–30) | 1 (0–6) | 0.006 |
| Overall time[#] | 3 (1–8) | 10 (3–24) | 25 (8–40) | 7 (2–24) | <0.001 |

[+] Reported as median (interquartile range).

[*] Kruskal-Wallis test.

[#] Overall time = Total duration of presentation and diagnosis time.

**Table 3. Sociodemographic and clinical factors associated with ≥3 months presentation delay in breast cancer patients.**

| Variable | <3 months presentation delay (%) | ≥3 months presentation delay (%) | Crude OR (95% CI) | p | Adjusted OR (95% CI) | p |
|---|---|---|---|---|---|---|
| Age (cont.) | - | - | 1.03 (0.99–1.07) | 0.137 | 1.02 (0.98–1.07) | 0.237 |
| Distance to health facility (cont.) | - | - | 1.00 (0.98–1.03) | 0.798 | - | - |
| Residence | | | | | | |
| Urban | 60 | 40 | Ref | | Ref | |
| Rural | 48.9 | 51.1 | 1.57 (0.78–3.17) | 0.210 | 1.55 (0.68–3.49) | 0.295 |
| Living Arrangement | | | | | | |
| With spouse only | 57.9 | 42.1 | Ref | | Ref | |
| With other than spouse | 62.5 | 37.5 | 0.83 (0.26–2.63) | 0.745 | 0.81 (0.23–2.82) | 0.734 |
| With spouse and other | 57 | 43 | 1.04 (0.38–2.82) | 0.942 | 1.46 (0.47–4.48) | 0.512 |
| Living alone | 16.7 | 83.3 | 6.88 (0.67–70.8) | 0.105 | 5.31 (0.45–63.0) | 0.186 |
| Education status[#] | | | | | | |
| ≥junior high school | 63.6 | 36.4 | Ref | | | |
| <junior high school | 46.8 | 53.2 | 1.99 (1.03–3.86) | 0.041 | - | - |
| Monthly income (IDR) | | | | | | |
| ≥3,000,000 | 69.5 | 30.5 | Ref | | Ref | |
| <3,000,000 | 48.3 | 51.7 | 2.43 (1.22–4.85) | 0.012 | 2.21 (0.99–4.93) | 0.052 |
| Consumption of herbal medicine before presentation | | | | | | |
| No | 56.8 | 43.2 | Ref | | Ref | |
| Yes | 56.6 | 43.4 | 0.99 (0.49–2.02) | 0.981 | - | - |
| Number of risk factor of BC known | | | | | | |
| ≥1 | 60.4 | 39.6 | Ref | | Ref | |
| None | 47.7 | 53.3 | 0.60 (0.30–1.22) | 0.156 | 0.98 (0.42–2.27) | 0.963 |
| Habit of breast self-exam | | | | | | |
| At least once a week | 54.8 | 45.2 | Ref | | Ref | |
| At least once a month | 61.1 | 38.9 | 0.77 (0.25–2.37) | 0.650 | - | - |
| Rarely or never | 56.7 | 43.3 | 0.93 (0.44–1.93) | 0.837 | - | - |
| First presenting symptom | | | | | | |
| Breast lump only | 57.4 | 42.6 | Ref | | Ref | |
| Breast lump and other | 42.9 | 57.1 | 1.80 (0.83–3.91) | 0.139 | 1.78 (0.78–4.06) | 0.174 |
| Other than breast lump | 85.7 | 14.3 | 0.23 (0.05–1.06) | 0.059 | 0.26 (0.05–1.32) | 0.104 |
| Number of comorbidities | | | | | | |
| None | 62.1 | 37.9 | Ref | | Ref | |
| 1 | 46 | 54 | 1.92 (0.95–3.88) | 0.068 | 1.80 (0.81–3.97) | 0.147 |
| ≥2 | 71.4 | 28.6 | 0.66 (0.23–1.91) | 0.440 | 0.63 (0.19–2.12) | 0.452 |
| Family history of cancer | | | | | | |
| No | 58.1 | 41.9 | Ref | | Ref | |
| Yes | 54.4 | 45.6 | 1.16 (0.60–2.26) | 0.659 | - | - |

Abbreviations: OR = Odds Ratio; CI = Confidence Interval; Ref = Reference; Cont. = continuous data; IDR = Indonesian Rupiah; BC = Breast Cancer.

#: not included in the multivariate analysis because it is highly correlated with monthly income.

-: not applicable.

Delays also increased the risk of having a lower BMI. The risk of BMI ≤23 at the time of diagnosis was significantly higher in participants with ≥3 months of delay in presentation (OR 2.08, 95% CI 1.07–4.01, p = 0.030) compared to those with no delays. Nevertheless, the risk was not significantly increased in participants with ≥1 month of delay in diagnosis when compared to their counterparts (OR 0.85, 95% CI 0.43–1.69, p = 0.644) (see S2 Table).

**Table 4. Sociodemographic, clinical factors and service utilization associated with ≥1 month diagnosis delay in breast cancer patients.**

| Variable | <1 month diagnosis delay (%) | ≥1 month diagnosis delay (%) | Crude OR (95% CI) | p | Adjusted OR (95% CI) | p |
|---|---|---|---|---|---|---|
| Age (cont.) | - | - | 0.98 (0.94–1.01) | 0.203 | 0.98 (0.93–1.02) | 0.286 |
| Residence | | | | | | |
| Urban | 33.3 | 66.7 | Ref | | | |
| Rural | 40 | 60 | 0.75 (0.36–1.54) | 0.434 | - | - |
| Living Arrangement | | | | | | |
| With spouse only | 31.6 | 68.4 | Ref | | | |
| With other than spouse | 31.2 | 68.8 | 1.02 (0.30–3.45) | 0.980 | - | - |
| With spouse and other | 36.6 | 63.4 | 0.80 (0.28–2.30) | 0.680 | - | - |
| Living alone | 50 | 50 | 0.46 (0.07–3.00) | 0.418 | - | - |
| Education status | | | | | | |
| ≥junior high school | 37.5 | 62.5 | Ref | | | |
| <junior high school | 32.3 | 67.7 | 1.26 (0.64–2.50) | 0.509 | - | - |
| Monthly income (IDR) | | | | | | |
| ≥3,000,000 | 35.6 | 64.4 | Ref | | | |
| <3,000,000 | 35.2 | 64.8 | 1.02 (0.51–2.02) | 0.957 | - | - |
| Frequency of medical visit before diagnosis | | | | | | |
| >3 times | 13.6 | 86.4 | Ref | | Ref | |
| ≤3 times | 49.4 | 50.6 | 0.16 (0.07–0.38) | <0.001 | 015 (0.06–0.37) | <0.001* |
| Types of health care facility first visited | | | | | | |
| Specialist (private or public) | 37.8 | 62.2 | Ref | | | |
| General practitioner (private or public) | 35.2 | 64.8 | 1.12 (0.53–2.35) | 0.772 | - | - |
| Midwife and other | 29.4 | 70.6 | 1.46 (0.44–4.86) | 0.540 | - | - |
| Number of risk factor of BC known | | | | | | |
| ≥1 | 32.1 | 67.9 | Ref | | Ref | |
| None | 43.2 | 56.8 | 1.61 (0.78–3.32) | 0.197 | 0.79 (0.34–1.84) | 0.586 |
| Habit of breast self-exam | | | | | | |
| At least once a week | 33.3 | 66.7 | Ref | | | |
| At least once a month | 33.3 | 66.7 | 1 (0.31–3.23) | 1.000 | - | - |
| Rarely or never | 36.7 | 63.3 | 0.86 (0.40–1.87) | 0.710 | - | - |
| First presenting symptom | | | | | | |
| Breast lump only | 37.6 | 62.4 | Ref | | Ref | |
| Breast lump and other | 25.7 | 74.3 | 1.74 (0.74–4.11) | 0.205 | 1.54 (0.59–4.02) | 0.378 |
| Other than breast lump | 42.9 | 57.1 | 0.80 (0.26–2.50) | 0.706 | 0.80 (0.23–2.81) | 0.727 |
| Number of comorbidities | | | | | | |
| None | 36.4 | 63.6 | Ref | | | |
| 1 | 33.3 | 66.7 | 1.14 (0.55–2.36) | 0.718 | - | - |
| ≥2 | 38.1 | 61.9 | 0.93 (0.34–2.56) | 0.886 | - | - |
| Family history of cancer | | | | | | |
| No | 39.8 | 60.2 | Ref | | Ref | |
| Yes | 28.1 | 71.9 | 1.69 (0.83–3.45) | 0.147 | 2.28 (1.03–5.04) | 0.042* |

Abbreviations: OR = Odds Ratio; CI = Confidence Interval; Ref = Reference; IDR = Indonesian Rupiah; BC = Breast Cancer.

*: statistically significant.

-: not applicable.

**Table 5. Effect of presentation and diagnosis delay on likelihood of stage III-IV breast cancer at point of diagnosis.**

| Presence of Delay | Likelihood of stage III-IV | | | | | |
|---|---|---|---|---|---|---|
| | Crude OR (95% CI) | p | Model 1 Adjusted OR (95% CI) | p | Model 2 Adjusted OR (95% CI) | p |
| **Presentation Delay** | | | | | | |
| Non-delay | Ref | | Ref | | Ref | |
| Delay (≥3 months) | 2.71 (1.29–5.72) | 0.009 | 2.30 (1.04–5.12) | 0.041 | 2.21 (0.97–5.01) | 0.059 |
| **Diagnosis Delay** | | | | | | |
| Non-delay | Ref | | Ref | | Ref | |
| Delay (≥1 months) | 3.25 (1.57–6.71) | 0.001 | 3.31 (1.42–7.72) | 0.006 | 3.03 (1.28–7.19) | 0.012 |

Abbreviations: OR = Odds Ratio; CI = Confidence Interval. Model 1 for presentation delay: adjusted for education level, monthly income, number of risk factors of breast cancer known, habit of breast self-exam, first presenting symptom, number of comorbidities; Model 1 for diagnosis delay: adjusted for education level, monthly income, frequency of medical visit before diagnosis, number of risk factor of breast cancer known, habit of breast self-exam, first presenting symptom, number of comorbidities; Model 2 for presentation delay: Model 1 for presentation delay + adjusted for diagnosis delay; Model 2 for diagnosis delay: Model 1 for diagnosis delay + adjusting for presentation delay.

## Reasons for presentation delay

Table 6 displayed reasons for 65 patients who waited for ≥3 months to seek help from health professionals after their first breast complaint. When asked why they did not go to a health facility earlier, the most frequent reason was not that the initial symptom did not cause bother and thinking that it was not a serious problem (27, 41.5%). This was followed by assuming that the symptoms were not cancer or a serious condition that required medical attention (18, 27.7%) and fear for surgery (17, 26.2%). For the whole cohort, there were two participants, one presented within 3 months and the other waited for ≥3 months, who stated that they were afraid of going out because of the possibility of contracting COVID-19 in the health facility (see S3 Table).

# Discussion

## Summary of key findings

This is the first quantitative study in Yogyakarta, Indonesia, of breast cancer cases determining the extent of delays in presentation and diagnosis exploring determinants with effect

**Table 6. Reasons provided by breast cancer patients for delays of ≥3 months' presentation (n = 65).**

| Reasons | Frequency (%) |
|---|---|
| The symptoms did not bother me/caused me pain. | 27 (41.5) |
| I thought it was not serious/cancer/did not require medical attention. | 18 (27.7) |
| I was afraid of undergoing surgery. | 17 (26.2) |
| I was too busy. | 9 (13.9) |
| I sought alternative treatment first. | 6 (9.2) |
| I was afraid of seeing a physician or going to a healthcare facility. | 5 (7.7) |
| I was afraid of the possible diagnosis. | 4 (6.2) |
| I was concerned about the cost. | 2 (3.1) |
| I was looking for a female physician. | 2 (3.1) |
| I needed someone to accompany me to the healthcare facility. | 1 (1.5) |
| I am embarrassed if my breast has to be examined. | 1 (1.5) |
| I was afraid to going out due to COVID-19 pandemic. | 1 (1.5) |

COVID-19 = Coronavirus Disease 2019.

estimation. Previous quantitative reports from Indonesia compared several factors for delays but did not determine the magnitude of identified differences [14, 15]. Being a region with the highest cancer prevalence in Indonesia, data on factors influencing delays is of importance to inform the targeting of interventions by national authorities. Delays in presentation were found for 43.3%, with delays potentially associated with a monthly household income of <3,000,000 IDR (equivalent to ~USD 200). Delays in diagnosis were experienced by 64.7% of participants, attributable to visiting a healthcare centre >3 times and family history of cancer. Our findings highlighted those women who presented late often presented with an advanced stage of the disease. This is consistent with extended data for various presentation and diagnosis times (see S2 Table). Moreover, the risk of lower BMI was also observed (see S3 Table), implying an increased cancer burden and reduced survival potential.

## Comparison of presentation and diagnosis delays with previous reports

The ≥3-month patient delay rate of 43.3% in the study population is higher than those observed in most high-income countries such as United States (17%) [7], Europe (17.3–20%) [6, 11] or Asia (29–35%) [9, 10]. It is comparable to reports from Asian countries including Malaysia and Iran (42.5–43.3%) [12, 29] and lower than other low- and middle-income countries like Pakistan (84%) [19] and Kenya (73.08%) [34]. A high proportion (64.7%) of ≥1 month diagnosis delay in our local breast cancer cohort is also higher than those previously documented (15.5–38%) [8, 11].

Factors that influence both delays in presentation and diagnosis have been extensively explored across international publications. However, their impact and mitigation strategies are yet to be described and evaluated in Indonesian settings. In our local setting, lower-income is potentially associated with a presentation delay, supporting findings in other countries [10, 19, 29]. The introduction of national universal health coverage in 2014 sought to increase access to healthcare to the population [35]. However, various out-of-pocket expenses remain, not covered by universal health coverage, such as transportation and logistics linked to presentation, diagnosis and subsequent treatments. Despite a large proportion of medical-related costs covered, there is still a financial burden placed on families with lower economic status. In addition, patients from lower socio-economic groups may have less health awareness and suboptimal family support [10].

Frequent (>3 times) medical visits after first initial consultation was the only significant predictor of delay in receiving a diagnosis in the present study. Although we explored patient-level factors, we did not specifically explore elements of the health service surrounding provider delays. The main factors for system-oriented delay include failure of medical practitioners to act on clinical findings, maltreatment of symptoms as benign breast disease, false-negative or misinterpretation of mammogram, and false-negative results of fine-needle aspiration cytology [9, 10, 12, 16, 36]. These factors may be targets for inclusion in future investigation and improvement initiatives. In the research literature, delays in diagnosis may be reduced through the provision of more efficient training programmes for members of the medical profession. Furthermore, our findings indicate a need for simpler and more efficient referral systems to access centres providing cancer care. As a response to long referral and waiting times in cancer care, the Swedish government launched a national policy called "standardized cancer patient pathways" [37]. This policy assigned all phases, from the first suspicion of cancer until the point of receiving treatment, with an appointed maximum time-scale, based on the optimal time for the patients and variation between diagnoses. Similar policies have been introduced across countries including Denmark and Norway [38, 39]. Furthermore, a family history of cancer is also associated with an increased risk of diagnosis delay in our

local cohort, supporting other reports [40, 41]. Fear of treatment and its side effects that may have been witnessed in other family members could be a reason for delaying diagnosis and requires further exploration.

Most patients tend to ignore symptoms, are not alarmed when they initially appear and do not limit daily activity. This may lead individuals to believe it is not harmful and thus delay seeking medical help. As much as 41.5% of women discredited the symptom due to the absence of pain and 27.7% thought it was not serious and thus did not require any medical attention. Delays in help-seeking behaviour have a major effect on a patient's prognosis and survival, yet, are potentially preventable. Recent studies reported a low awareness level of breast cancer risk factors, barriers, attitudes and breast cancer screening among Indonesian women. This may reflect an inadequate or a lack of breast cancer awareness in the country [42, 43]. Additional education programs aiming to increase awareness and public education has been recommended [43] to improve awareness of breast cancer signs and symptoms. There is an urgent need to develop communication and education strategies regarding breast cancer symptoms and early detection for Indonesia, such as specifically motivating early detection practices and breast self-examination [44]. Various methods could be explored, including mass media, various forms of literature, and programmes. One example is the 'Be Cancer Alert Campaign', using mass media for raising breast and colorectal cancer awareness conducted in Malaysia [45]. In one systematic review, a breast cancer awareness campaign was found to increase the initiation of breast self-examination behaviours and increase the attendance of breast cancer screening [46]. There is scope to explore the feasibility and wider evaluation of similar campaigns in Indonesia. For example, a policy of breast cancer self-examination was introduced by the Indonesian Ministry of Health in 2015 but has yet to be comprehensively evaluated. Our findings reemphasize the need for cancer prevention programs to focus on making women aware of any symptoms in the breast, especially in those with lower economic status. Others indicated that women living in urban areas have a poorer level of knowledge of breast cancer risk factors compared to those living in more rural areas. This group may also serve as a target for future awareness programs [43].

During this study, reluctance to go to attend a health facility during the COVID-19 pandemic was a reason for delayed presentation in one case. Among 65 patients who delayed their first consultation, 10 initially presented to the health facility at the time of the COVID-19 pandemic in Indonesia. When considering the whole panel (n = 150), 2 cases out of 45 women stated the same reason. There was an indication that this became a reason for common delays during the study period. Its effect on the overall health problem in Indonesia still needs to be evaluated.

## Study strength and limitations

We gathered detailed data from participants in an existing cohort study, using a validated questionnaire that was used by trained researchers in ambulatory clinics. Inclusion from the existing cohort study may introduce some limitations. The study mostly recruited women with good ECOG index, without terminal disease, and who were chemotherapy-naive. Some patients have died because of a terminal disease that developed after diagnosis and treatment administration. Patients with very poor clinical performance and heavily pretreated cases were not included due to the exclusion criteria of the main study. Furthermore, the vast majority of patients underwent staging in the hospital site in which the research team is based (type A hospital) so that there were time intervals between diagnostic confirmation in the district hospitals (type B hospitals) and stage establishment. Some cases possibly have already developed a more advanced stage during the elapsing time. However, a 1-month interval is indicated as

representative staging time after being diagnosed in the previous health care [47]. Those who did not experience diagnosis delay in our cohort had a median referral time interval from type B hospitals to ours of 54 days. Many interviews carried out in this study were completed months after participant recruitment to the main study. The first case was recruited in May 2018 while interviews began in September 2020. As a consequence, there is a 1–29 month interval between the first intake in the main study and interviews for this study. Although strategies were in place to mitigate recall bias, this may still have influenced participant responses.

## Conclusions

Many breast cancer patients in our local setting delayed seeking advice for symptoms later diagnosed as breast cancer symptoms. A high proportion of women experienced a delay in diagnosis. Delays significantly increased the risk of presentation with advanced disease and its association with high mortality probability. Frequent medical visits before diagnosis and family history of cancer were significant determinants of diagnosis delays. Feasibility testing of approaches to promoting community education to promote breast cancer awareness and training for health care professionals is required to explore strategies for potentially minimising delays and mortality from breast cancer in our local region and other settings across Indonesia.

## Supporting information

**S1 File. Questionnaire in English version.**
(DOC)

**S2 File. Questionnaire in Bahasa Indonesia version.**
(DOC)

**S1 Data set. Minimal data set.**
(XLSX)

**S1 Fig. Steps for developing the study questionnaire.** Questionnaire development started from identifying variables that will be measured and items that were obtained from existing questions related to the selected variables. It is followed by item selection and translation. The last steps included validity assessment or face validity and questionnaire finalization.
(TIF)

**S1 Table. Proportion of presentation and diagnosis time based on stage at diagnosis.**
(DOCX)

**S2 Table. Effect of presentation and diagnosis delay on likelihood of lower BMI ($<$23) at point of diagnosis.**
(DOCX)

**S3 Table. Reasons provided across all study participants for presentation delay (n = 150).**
(DOCX)

## Acknowledgments

We gratefully thanked Irfan Haris, Riani Witaningrum, Yufi Kartika Astari, and Betrix Rifana Kusumaning Indah for technical assistance.

## Author Contributions

**Conceptualization:** Susanna Hilda Hutajulu, Yayi Suryo Prabandari, Matthew John Allsop.

**Data curation:** Susanna Hilda Hutajulu, Yayi Suryo Prabandari, Bagas Suryo Bintoro, Juan Adrian Wiranata, Mentari Widiastuti, Norma Dewi Suryani, Rorenz Geraldi Saptari, Matthew John Allsop.

**Formal analysis:** Susanna Hilda Hutajulu, Yayi Suryo Prabandari, Bagas Suryo Bintoro, Juan Adrian Wiranata, Mentari Widiastuti.

**Funding acquisition:** Susanna Hilda Hutajulu, Matthew John Allsop.

**Investigation:** Susanna Hilda Hutajulu, Yayi Suryo Prabandari, Juan Adrian Wiranata, Norma Dewi Suryani, Rorenz Geraldi Saptari, Kartika Widayati Taroeno-Hariadi, Johan Kurnianda, Ibnu Purwanto, Mardiah Suci Hardianti.

**Methodology:** Susanna Hilda Hutajulu, Yayi Suryo Prabandari, Bagas Suryo Bintoro, Juan Adrian Wiranata, Mentari Widiastuti, Matthew John Allsop.

**Project administration:** Susanna Hilda Hutajulu, Juan Adrian Wiranata, Norma Dewi Suryani.

**Resources:** Susanna Hilda Hutajulu, Yayi Suryo Prabandari, Kartika Widayati Taroeno-Hariadi, Johan Kurnianda, Ibnu Purwanto, Mardiah Suci Hardianti.

**Software:** Bagas Suryo Bintoro, Juan Adrian Wiranata, Mentari Widiastuti.

**Supervision:** Susanna Hilda Hutajulu, Yayi Suryo Prabandari, Kartika Widayati Taroeno-Hariadi, Johan Kurnianda, Ibnu Purwanto, Mardiah Suci Hardianti.

**Validation:** Susanna Hilda Hutajulu, Yayi Suryo Prabandari, Juan Adrian Wiranata, Mentari Widiastuti, Norma Dewi Suryani, Rorenz Geraldi Saptari, Johan Kurnianda, Ibnu Purwanto, Mardiah Suci Hardianti, Matthew John Allsop.

**Visualization:** Susanna Hilda Hutajulu, Yayi Suryo Prabandari, Juan Adrian Wiranata, Matthew John Allsop.

**Writing – original draft:** Susanna Hilda Hutajulu, Yayi Suryo Prabandari, Juan Adrian Wiranata, Rorenz Geraldi Saptari.

**Writing – review & editing:** Susanna Hilda Hutajulu, Yayi Suryo Prabandari, Bagas Suryo Bintoro, Juan Adrian Wiranata, Mentari Widiastuti, Mardiah Suci Hardianti, Matthew John Allsop.

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
