## [Decision Letter · Decision Letter 0]

28 Jul 2021

PONE-D-21-15358

Delays in the presentation and diagnosis of women with breast cancer in Yogyakarta, Indonesia: a retrospective observational study

PLOS ONE

Dear Dr. Hutajulu,

Thank you for submitting your manuscript to PLOS ONE. After careful consideration, we feel that it has merit but does not fully meet PLOS ONE’s publication criteria as it currently stands. Therefore, we invite you to submit a revised version of the manuscript that addresses the points raised during the review process.

We look forward to receiving your revised manuscript.

Kind regards,

Evy Yunihastuti, MD

Academic Editor

PLOS ONE

6. Please note that in order to use the direct billing option the corresponding author must be affiliated with the chosen institute. Please either amend your manuscript to change the affiliation or corresponding author, or email us at plosone@plos.org with a request to remove this option.

Additional Editor Comments (if provided):

Reviewers' comments:

Reviewer's Responses to Questions

**Comments to the Author**

1. Is the manuscript technically sound, and do the data support the conclusions?

Reviewer #1: Yes

Reviewer #2: Yes

2. Has the statistical analysis been performed appropriately and rigorously? 

Reviewer #1: No

Reviewer #2: No

3. Have the authors made all data underlying the findings in their manuscript fully available?

Reviewer #1: Yes

Reviewer #2: Yes

4. Is the manuscript presented in an intelligible fashion and written in standard English?

Reviewer #1: No

Reviewer #2: Yes

5. Review Comments to the Author

Reviewer #1: Major comments

Line 167: you have stated that you have done content validity, but not described the result of the validity test. so explain the results of the validity test at the beginning of your result part

line 170-172: specify the exact number of respondents at the Ambulatory clinic and delivered through phone call.

Line 185: have you used any methods that can help respondents to recall their exact date of their first symptom? if so please describe it here.

Line 222: Describe the BMI classification in detail. what do you mean by (underweight to normal) and (normal to overweight)?

Line 235: why do you use Kruskal Wallis test? have you checked any assumptions not to do one way ANOVA? please explain it in your result part if you have checked the assumptions, if not it will not be the appropriate statistical analysis.

besides, you have already transform the data in to categorical variable which doesn't allow you to do One way ANOVA or Kruskal Wallis test (only applied for continuous data). so justify you reasons

Line 271: the table titles should be self explanatory which answers "what, who, where, and when", So apply it for all of your Table titles

On table 1: second column write "Frequency n (%)"

Age: Mean should not be explained alone, so write "mean (SD)"

Comorbidities: you should list the commonest comorbidities with their frequency not the number of comorbidities

Table 2: it is not the appropriate way of analysis of a Kruskal Wallis test, please make it self explanatory, the numbers in each cell are not understandable.

Table 3: this association doesn't tell us the exact relationship of stage at diagnosis and presentation delay because it is just crude odds ratio. to know their relatuion stage of diagnosis should be entered to the model and see the effect

Table 4: second column should not be put alone. so add another column and show the 'cross tab' of all independent variables with both "<3 months and >3 months" so that it will be explanatory.

Table 5: Apply the comments above on table 4

Line 537: References should be updated, Please use references that are published since 2010

Minor comments

Line 34: remove repeated words: "Extent of, impact of" Just say "factors associated with"

Line 36: "Cross-sectional study nested in an ongoing prospective cohort study of BC patients". it is not a full statement Add " were employed" after the last word

Line 78: correct grammar and English flow

Line 113: Remove the word Impact and use associated factors through out the paper

Generally the English is good but needs revision to make easily readable and understandable for the readers

Reviewer #2: Major issues:

1. Questionnaire development.

It is not clearly define in the method, how to avoid recall bias that might influence the results of the study (page 38, line 479-481).

2. The analysis (multivariable logistic regression) (comments to the question no.2)

a. The authors need to clarify the reason of choosing ‘living alone’ (in living arrangement variable) as reference in multivariable logistic regression analysis (Table 4). If ‘living alone’ is considered as the highest risk among other variables (with spouse only, with spouse and other, or with other than spouse), in my opinion, the reference should be a variable that has the lowest risk associated with > 3 months presentation delay.

b. The same reason of choosing the ‘luminal A’ as the reference to luminal B, Her2- enriched and TNBC variables, considering that each variable has different molecular and clinical characteristic. (Table 4)

c. The same reason of choosing the ‘general practitioner’ (in health care facility first visited) as the reference to specialist and midwife variables that associated with >1 month diagnosis delay, assuming that specialist is the ‘best’ or the lowest risk health care facility to avoid delay in diagnosis. (Table 5)

3. It is interesting that the authors analyze of the molecular type of the breast cancer (luminal A, luminal B, Her2-enriched and TNBC) with the presentation delays (Table 4). However, it is not clearly define in the literatures whether the molecular type is associated the diagnosis delays. Most of the delays in diagnosis process related to the stage of the disease, and the molecular types is usually associated with the prognosis of the disease.

Minor issue:

The authors provide a good point of discussion, however the use of word ‘cohort’ (page 34, line 374) is not suitable with the design study (cross sectional).

6. PLOS authors have the option to publish the peer review history of their article (what does this mean?). If published, this will include your full peer review and any attached files.

Reviewer #1: No

Reviewer #2: No

---

## [Author Response · Author response to Decision Letter 0]

11 Sep 2021

Dear editor and reviewers,

We are thankful for the positive feedback received from reviewers 1 and 2, and the editorial team, and for the opportunity to respond to the constructive points in our accompanying revised manuscript. Please find below our point-by-point response to reviewer feedback outlining, where relevant, the related changes we have made. We have uploaded revised versions of the manuscript as instructed, including both a clean copy and a track changes version. 

Editor’s comment

Response:

We have checked these requirements ahead of submitting our revised manuscript.

Response:

We have included as Supporting Information a copy of questionnaire that we used in this study both in Bahasa Indonesia and English.

Response:

Thank you for your remarks. We have included a statement outlining the funders and ensured they match in both the ‘Funding Information’ and ‘Financial Disclosure’ sections. We have not included specific grant reference numbers as these were funds awarded to the respective institutions that did not provide specific reference numbers in allocating the funds for this research.

Response:

We have provided as Supporting Information the minimum dataset that were used to generate Fig 1. Fig 1 is the figure that reflects the main finding presented in the paper. Sharing of the full de-identified dataset is not possible due to restrictions imposed by the ethics committee as most of these contain patient data, albeit de-identified, and it may be possible to determine the identity of participants given the extent of sociodemographic and clinical data available for each participant. Should there be a request for data, this can be sent to the corresponding author (email: susanna.hutajulu@ugm.ac.id). Future researchers can contact the institutional ethics committee (email: mhrec_fmugm@ugm.ac.id) at Universitas Gadjah Mada, Indonesia, with data access queries as well. 

We have also included our response to this comment in the accompanying cover letter. 

Response:

Thank you for your comment. Indeed, we provided two ethics statements for both the parent and the present study. Both statements have been combined into one section of text and are now presented together under Methods (now page 9, lines 183-188).

6. Please note that in order to use the direct billing option the corresponding author must be affiliated with the chosen institute. Please either amend your manuscript to change the affiliation or corresponding author, or email us at plosone@plos.org with a request to remove this option.

Response:

I sent an email to plosone@plos.org with a request to remove this direct billing option.

Reviewer #1: Major comments

1. Line 167: you have stated that you have done content validity, but not described the result of the validity test. so explain the results of the validity test at the beginning of your result part

Response:

Thank you for highlighting this issue. A comprehensive overview of the questionnaire development was included on pages 7-9, lines 139-171. We conducted a rigorous development process which resulted in a set of 35 questions. However, we did not conduct a quantitative content validity and instead focused on the face validity and readability of questionnaire which were reviewed qualitatively. We have now removed mention of content validity from the manuscript. Our development process focused on making sure the questions were meaningful and easy to understand to gather data in the absence of validated approaches for Indonesia. We have outlined our approach to assessing the face validity in the methods section. We have also clarified the description of the final number of items contained in the questionnaire from 21 to 35. 21 is the number of items we developed at the earliest phase of our study.

2. Line 170-172: specify the exact number of respondents at the Ambulatory clinic and delivered through phone call.

Response:

In the manuscript text (previous lines 170-172, now lines 180-182) we have added the exact number of respondents both at the ambulatory clinic (n=77) and delivered through phone call (n=74).

3. Line 185: have you used any methods that can help respondents to recall their exact date of their first symptom? if so please describe it here.

Response:

Thank you for your question about methods used to support respondents to recall the date of their first symptom. When respondents could not remember the date of when the symptoms first appeared or the date when they first visited a health facility, we asked for a time span in months. After providing the range of months, we sought to direct the patient to remember the distance between those dates from important dates or events such as a family or respondent’s birthday, or religious holidays, to further narrow the range and improve the recall closer to the exact date. When provided with a single month, we tried to ask about the exact day or its distance with other important events in the month, in order to further narrow the date into a single exact day. We also drew on breast surgery and procedure dates that were recorded in clinical notes as a benchmark, because it was considered by most patients as an important event. 

We have incorporated further details about this approach into the manuscript text in the Methods (Methods - Delays definition) (lines 195-206).

This response also applies to the question raised by Reviewer 2 relating to strategies for overcoming recall bias.

4. Line 222: Describe the BMI classification in detail. what do you mean by (underweight to normal) and (normal to overweight)?

Response:

We used the World Health Organization Asia-Pacific body mass index classifications, which classified BMI into underweight (<18.5), normal (18.5-22.9), overweight (23-24.9) and obese (≥25). In our analysis, we further stratified BMI into two groups: underweight to normal (<23) and overweight to obese (≥23). We adjusted the corresponding text in the manuscript to provide further clarity around these groupings of BMI scores (previous lines 222-223, now lines 244-248) and tables (Table 1 and S2 Table).

5. Line 235: why do you use Kruskal Wallis test? have you checked any assumptions not to do one way ANOVA? please explain it in your result part if you have checked the assumptions, if not it will not be the appropriate statistical analysis.

besides, you have already transform the data in to categorical variable which doesn't allow you to do One way ANOVA or Kruskal Wallis test (only applied for continuous data). so justify you reasons

Response:

Many thanks for your feedback. In line with our statement in statistical analyses (line 235, now line 260), “Kruskal-Wallis statistical test was performed to assess the length of presentation and diagnosis delay among the patients and distribution of delays by stage at presentation”. 

We visually inspected the distribution of the data using the histogram. The histograms on diagnosis time and presentation time showed that the data of both variables are skewed to the right. Due to this violation of normality, we could not use ANOVA test (reference: Hu, Z. D., Zhou, Z. R., & Qian, S. (2015). How to analyze tumor stage data in clinical research. Journal of thoracic disease, 7(4), 566–575. https://doi.org/10.3978/j.issn.2072-1439.2015.04.09).

We have added the following text to outline our rationale for using the Kruskall Wallis test: “This analysis was chosen as data were not normally distributed. We visually inspected the distribution of the data using a histogram. The histogram on diagnosis time and presentation time showed that the data of both variables were skewed to the right” (lines 262-273). We have included a histogram of the data, below, for the reviewer’s information. 

In terms of our transformation of data, as outlined in Table 2, we used the data in its original (continuous) form. We presented the median of the delays in each stage and conducted our analysis using the Kruskal Wallis test. The data differed from those in Table 1 that were already transformed to categorical variables.

6. Line 271: the table titles should be self explanatory which answers "what, who, where, and when", So apply it for all of your Table titles

Response:

Thank you very much for your suggestion. We modified all table titles to make them more self-explanatory. Related to tables, we have reordered their appearance in the findings section to accommodate a better flow with reporting our findings. For information, Table 3 became Table 5, Table 4 became Table 3 and Table 5 became Table 3. 

The table titles have been outlined as below, showing the original and revised titles: 

Table 1. Characteristics of study subjects (n =150)

New title: Sociodemographic, clinical, and delay characteristics of study participants (n =150) recruited from Dr. Sardjito Hospital, Yogyakarta, in 2018-2021.

Table 2. Median of presentation and diagnosis time by stage at baseline.

New title: Median presentation, diagnosis, and overall delay for all participants (n=150) by stage at diagnosis.

Table 3 � became Table 5. Delays and risk of more advance stage at time of diagnosis.

New title: Effect of presentation and diagnosis delay on likelihood of stage III-IV breast cancer at point of diagnosis.

Table 4 � became Table 3. Factors associated with ≥3 months presentation delay.

New title: Sociodemographic and clinical factors associated with ≥3 months presentation delay in breast cancer patients.

Table 5 � became Table 4. Factors associated with ≥1 month diagnosis delay.

New title: Sociodemographic, clinical factors and service utilization associated with ≥1 month diagnosis delay in breast cancer patients.

Table 6. Reasons for presentation delay (n=65).

New title: Reasons provided by breast cancer patients for delays of ≥3 months’ presentation (n=65).

S1 Table. Proportion of presentation and diagnosis time based on stage.

New title: Proportion of presentation and diagnosis time based on stage at diagnosis.

S2 Table. Delays and risk of lower BMI at time of diagnosis.

New title: Effect of presentation and diagnosis delay on likelihood of lower BMI (<23) at point of diagnosis.

S3 Table. Reasons for presentation delay in the whole panel (n=150).

New title: Reasons provided across all study participants for presentation delay (n=150).

7. On table 1: second column write "Frequency n (%)"

Age: Mean should not be explained alone, so write "mean (SD)"

Comorbidities: you should list the commonest comorbidities with their frequency not the number of comorbidities

Response:

Thank you. We have revised the manuscript in line with this suggestion. In Table 1, we have placed frequency and n (%) in the heading of second column. We have also reported standard deviation alongside the mean age of the study population. 

In terms of comorbidities, we have now added specific information about condition and their frequency to Table 1. For information, the total frequency for comorbidities is >100% as participants may have reported more than 1 of the conditions reported.

8. Table 2: it is not the appropriate way of analysis of a Kruskal Wallis test, please make it self explanatory, the numbers in each cell are not understandable.

Response:

Thank you for your remarks. Our response about the rationale for the Kruskall Wallis statistical test is outlined above (Response to question no 5). In Table 2, to improve the readability of the table and ensure it is self-explanatory, we have modified title and headings. We have also added lines to group the analyses presented, making the comparison clearer of median delay across each category (presentation, diagnosis, overall) by stage of disease.

9. Table 3: this association doesn't tell us the exact relationship of stage at diagnosis and presentation delay because it is just crude odds ratio. to know their relatuion stage of diagnosis should be entered to the model and see the effect.

Response:

Thank you for this comment. In response to this comment, we modified Table 3 in the following ways:

• We reordered Table 3 into Table 5 to show factors associated with delays first. Now Table 3 is for factors associated with presentation delay and Table 4 is for factors associated with diagnosis delay.

• We now show in Table 5 the effect of delays to diagnosis at first breast cancer diagnosis in two models. 

• Model 1 for presentation delay: adjusted for education level, monthly income, number of risk factor of breast cancer known, habit of breast self-exam, first presenting symptom, number of comorbidities, molecular subtype; 

• Model 1 for diagnosis delay: adjusted for education level, monthly income, frequency of medical visit before diagnosis, number of risk factor of breast cancer known, habit of breast self-exam, first presenting symptom, number of comorbidities, molecular subtype

• Model 2 for presentation delay: Model 1 for presentation delay + adjusted for diagnosis delay.

• Model 2 for diagnosis delay: Model 1 for diagnosis delay + adjusting for presentation delay. 

10. Table 4: second column should not be put alone. so add another column and show the 'cross tab' of all independent variables with both "<3 months and >3 months" so that it will be explanatory.

Response:

We have modified Table 4 (now Table 3) by adding another column to show variables with <3 months alongside those with ≥3 months presentation delay to make it more explanatory.

11. Table 5: Apply the comments above on table 4

Response:

We have modified Table 5 (now Table 4) by adding another column to show variables with <1 month along with ≥1 month diagnosis delay to make it more explanatory.

12. Line 537: References should be updated, Please use references that are published since 2010

Response:

Thank you for the suggestion. We have reviewed the references closely. Where possible, we have revised references used, but also outline where references remain due to, for example, a lack of other published data for Indonesia. 

References that were published before 2010 and have been removed or replaced include: 

Ref [6] (Arndt, 2006) 

Ref [7] (Meechan, 2003) 

Ref [8] (Velikova, 2004) 

Ref [9] (Jenner, 2000) 

Ref [10] (Nosarti, 2000) 

Ref [11] (Montella, 2001) 

Ref [17] (Montazeri, 2003) 

Ref [18] (Pineros, 2009) 

Ref [33] (Cancer Research UK, 2009)

Ref [36] (Facione, 1993) 

Ref [37] (Richards, 1999) 

Ref [58] (Neale, 1986)

References that remain include Ref [4] (Wahyuni, 2002), because is it the only data from Indonesia showing 5-year overall survival rate data of breast cancer in Indonesia. We keep Ref [25] (Bish, 2005) because it is the most recent literature we could find when determining our decision of choosing ≥1 month to define diagnosis delay in our study. We keep Ref [34] (Harirchi, 2005) because it provides fundamentals that underpin questionnaire development. Lastly, our citation for Braun and Clarke (2006) [42] is the leading citation for thematic analysis and its use in research, which remains in the manuscript. 

We also added more recently published literature, including Cancer Research UK 2011, Brzozowska et al, 2014, Li et al, 2019, Scheel et al, 2020, Albeshan et al, 2020, Kim et al 2012, and Caplan 2014 to complement the introduction and discussion regarding questionnair variables, presentation and diagnosis delay, and explanation of molecular subtypes in the multivariable model. 

Reviewer #1: Minor issues

1. Line 34: remove repeated words: "Extent of, impact of" Just say "factors associated with"

Response:

We have edited the manuscript in line 34 as suggested. We have also updated other sections of the manuscript to reflect these changes, including in the objective.

2. Line 36: "Cross-sectional study nested in an ongoing prospective cohort study of BC patients". it is not a full statement Add " were employed" after the last word

Response:

We also have edited the manuscript in line 36 as suggested: “A cross-sectional study nested in an ongoing prospective cohort study of breast cancer patients at Dr Sardjito Hospital, Yogyakarta, Indonesia, was employed.”

3. Line 78: correct grammar and English flow

Response:

We modified the sentence in line 78 (now line 77) to “Delay in breast cancer presentation and diagnosis are likely to be key factors in advanced-stage presentation. There are disparities in the length of delays between countries.”

4. Line 113: Remove the word Impact and use associated factors through out the paper

Response:

We modified the sentence in lines 112-115 (now lines 113-115) to “The objective of this study is to quantitatively investigate the factors associated with presentation and diagnosis delays, relationship of delays to stage at presentation and reasons for patient delay within local breast cancer cases.”

Reviewer #2: Major issues:

1. Questionnaire development.

It is not clearly define in the method, how to avoid recall bias that might influence the results of the study (page 38, line 479-481).

Response:

We have outlined our approach to avoiding recall bias as part of our response to comments from Reviewer 1 (Response to question no 3). Please see above for further details.

2. The analysis (multivariable logistic regression) (comments to the question no.2)

a. The authors need to clarify the reason of choosing ‘living alone’ (in living arrangement variable) as reference in multivariable logistic regression analysis (Table 4). If ‘living alone’ is considered as the highest risk among other variables (with spouse only, with spouse and other, or with other than spouse), in my opinion, the reference should be a variable that has the lowest risk associated with > 3 months presentation delay.

b. The same reason of choosing the ‘luminal A’ as the reference to luminal B, Her2- enriched and TNBC variables, considering that each variable has different molecular and clinical characteristic. (Table 4)

c. The same reason of choosing the ‘general practitioner’ (in health care facility first visited) as the reference to specialist and midwife variables that associated with >1 month diagnosis delay, assuming that specialist is the ‘best’ or the lowest risk health care facility to avoid delay in diagnosis. (Table 5)

Response:

Thank you for your comment. 

In response to a, in Table 4 (now Table 3), we used “living alone” as a reference due to its highest risk associated with > 3 months presentation delay. 

In response to b, we used luminal as a reference of the lowest risk based on previous report (ref: Galukande et al. 2014). 

In response to c, general practitioner is a reference of the highest risk associated with > 3 months presentation delay based on report from previous studies. In the analyses of types of health care facility first visited we put midwife as the highest, based on our assumption since there is no literature including midwife in the model. 

We have considered your comments and have now set all determinants with lowest risk stratification for outcome as reference in the logistic regression analyses in Table 4 and 5 (now Table 3 and 4). These tables now reflect the updated analyses, with the reference points modified as suggested.

3. It is interesting that the authors analyze of the molecular type of the breast cancer (luminal A, luminal B, Her2-enriched and TNBC) with the presentation delays (Table 4). However, it is not clearly define in the literatures whether the molecular type is associated the diagnosis delays. Most of the delays in diagnosis process related to the stage of the disease, and the molecular types is usually associated with the prognosis of the disease.

Response:

It is true that molecular types have been long observed as a prognosis factor for breast cancer. The role and association of molecular types with clinical symptom recognition has also been raised by others, too (Galukande 2014 [20], Kim 2012 [39]). It is, in fact, an area in which there is limited evidence, but we are keen to explore its relevance to self-detection in our breast cancer population given limited literature and the availability of data on molecular type available for the study population. We used luminal A as reference for analyses based on the result of Galukande 2014 showing cases with TNBC and HER2 tumors as late presenters. This is supported by other research (Kim 2012) that observed luminal A tumors to be the most symptomatic. In contrast, our study observed TNBC cases as earlier presenters compared with cases with luminal A tumors, although significance was not reached. This conflicting result reflects the early stage of evidence in this area and highlights the need for further work to explore how molecular types manifest as symptoms and their subsequent impact on outcomes that may influence presentation, such as help-seeking behaviours.

Reviewer #2: Minor issue:

1. The authors provide a good point of discussion, however the use of word ‘cohort’ (page 34, line 374) is not suitable with the design study (cross sectional).

Response:

We have edited the manuscript as suggested in line 374 (now line 424) replacing ‘cohort’ with ‘study population’: “The ≥3-month patient delay rate of 43.3% in the study population is higher than those observed in most high-income countries.”

---

## [Decision Letter · Decision Letter 1]

25 Oct 2021

PONE-D-21-15358R1Delays in the presentation and diagnosis of women with breast cancer in Yogyakarta, Indonesia : a retrospective observational studyPLOS ONE

Dear Dr. Hutajulu

Thank you for submitting your manuscript to PLOS ONE. After careful consideration, we feel that it has merit but does not fully meet PLOS ONE’s publication criteria as it currently stands. Therefore, we invite you to submit a revised version of the manuscript that addresses the points raised during the review process.

 Do follow the acceptable repositories as recommended by Plos One, not just dataset to generate figure 1, but also table 1-6. Please see http://journals.plos.org/plosone/s/data-availability#loc-recommended-repositories. Please submit your revised manuscript by Dec 09 2021 11:59PM. If you will need more time than this to complete your revisions, please reply to this message or contact the journal office at plosone@plos.org. Please include the following items when submitting your revised manuscript:A rebuttal letter that responds to each point raised by the academic editor and reviewer(s). You should upload this letter as a separate file labeled 'Response to Reviewers'.A marked-up copy of your manuscript that highlights changes made to the original version. You should upload this as a separate file labeled 'Revised Manuscript with Track Changes'.An unmarked version of your revised paper without tracked changes. You should upload this as a separate file labeled 'Manuscript'.If applicable, we recommend that you deposit your laboratory protocols in protocols.io to enhance the reproducibility of your results. Protocols.io assigns your protocol its own identifier (DOI) so that it can be cited independently in the future. For instructions see: https://journals.plos.org/plosone/s/submission-guidelines#loc-laboratory-protocols. Additionally, PLOS ONE offers an option for publishing peer-reviewed Lab Protocol articles, which describe protocols hosted on protocols.io. Read more information on sharing protocols at https://plos.org/protocols?utm_medium=editorial-email&utm_source=authorletters&utm_campaign=protocols.

We look forward to receiving your revised manuscript.

Kind regards,

Evy Yunihastuti, MD

Academic Editor

PLOS ONE

Journal Requirements:

Reviewers' comments:

Reviewer's Responses to Questions

**Comments to the Author**

1. If the authors have adequately addressed your comments raised in a previous round of review and you feel that this manuscript is now acceptable for publication, you may indicate that here to bypass the “Comments to the Author” section, enter your conflict of interest statement in the “Confidential to Editor” section, and submit your "Accept" recommendation.

Reviewer #1: (No Response)

Reviewer #2: (No Response)

2. Is the manuscript technically sound, and do the data support the conclusions?

Reviewer #1: Yes

Reviewer #2: Partly

3. Has the statistical analysis been performed appropriately and rigorously? 

Reviewer #1: Yes

Reviewer #2: Yes

4. Have the authors made all data underlying the findings in their manuscript fully available?

Reviewer #1: Yes

Reviewer #2: Yes

5. Is the manuscript presented in an intelligible fashion and written in standard English?

Reviewer #1: Yes

Reviewer #2: Yes

6. Review Comments to the Author

Reviewer #1: (No Response)

Reviewer #2: 

Thank you for the opportunity to review the manuscript “Delays in the presentation and diagnosis of women of breast cancer in Yogyakarta, Indonesia: a retrospective observational study”. Please find my review of the revised manuscript below.

Major issues:

1. Questionnaire development (recall bias)

The authors explained the approach to avoid recall bias as part of their response to Reviewer #1.

2. The analysis (multivariable logistic regression)

The authors made revision and updated the analyses as well as the Tables (Table 3 and 4).

3. The authors explained the reason of the analysis of molecular type of breast cancer with diagnosis delay (Tables 3 and 4). However, the literature used by the authors (Galukande 2014) is a cross-sectional study to determine the prevalence of breast cancer molecular phenotypes. In my opinion, this reference is not suitable with the purpose of the analysis made by the authors .

Molecular type is an immunohistochemical classification or characterization of breast cancer for hormone receptor status or HER-2 gene overexpression and it correlates with the clinical outcome or the choice of systemic therapy (e.g. ESMO Educational sessions in Annals of Oncology vol 23 suppl 10, Sept 1,2012; National Comprehensive Cancer Network Guidelines version 4.2020; or Nature/NPJ Breast Cancer 6 March 2020). Molecular types also correlate with prognosis of breast cancer (2019 ASCO education books - Molecular testing in breast cancer by Jennifer K. Litton et al). Unfortunately, there is no relevant literature of molecular type of breast cancer in association with diagnosis delays. I am concern that this issue is the major issue of the manuscript.

Minor issue:

The authors edited the design as suggested.

7. PLOS authors have the option to publish the peer review history of their article (what does this mean?). If published, this will include your full peer review and any attached files.

Reviewer #1: No

Reviewer #2: No

---

## [Author Response · Author response to Decision Letter 1]

12 Nov 2021

Dear editor and reviewers,

We are thankful for the positive feedback received from Reviewers 1 and 2, and the editorial team, and for the opportunity to respond to the constructive points in our accompanying revised manuscript. Please find below our point-by-point response to reviewer feedback outlining, where relevant, the related changes we have made. We have uploaded revised versions of the manuscript as instructed, including both a clean copy and a track changes version. 

Editor comments

1. Do follow the acceptable repositories as recommended by Plos One, not just dataset to generate figure 1, but also table 1-6. Please see http://journals.plos.org/plosone/s/data-availability#loc-recommended-repositories.

Response:

We have revised the dataset to now include deidentified data used to generate Figure 1 and Tables 1-6. We have provided derived values for date information to avoid identification of participants. Sharing of the full de-identified dataset is not possible due to restrictions imposed by the ethics committee as most of these contain patient data, albeit de-identified, and it may be possible to determine the identity of participants given the extent of sociodemographic and clinical data available for each participant. 

Should there be a request for data, this can be sent to the corresponding author (email: susanna.hutajulu@ugm.ac.id). Future researchers can contact the institutional ethics committee (email: mhrec_fmugm@ugm.ac.id) at Universitas Gadjah Mada, Indonesia with data access queries as well. 

We have included again our response to this comment in the accompanying cover letter. 

Response:

Not applicable.

Response:

Thank you for your comment. We have checked all references and found no article with a retraction notice.

Reviewer 2

Major issues

1. Questionnaire development (recall bias)

The authors explained the approach to avoid recall bias as part of their response to Reviewer #1.

Response: Thank you for your approval.

2. The analysis (multivariable logistic regression)

The authors made revision and updated the analyses as well as the Tables (Table 3 and 4).

Response:

Thank you for your approval.

3. The authors explained the reason of the analysis of molecular type of breast cancer with diagnosis delay (Tables 3 and 4). However, the literature used by the authors (Galukande 2014) is a cross-sectional study to determine the prevalence of breast cancer molecular phenotypes. In my opinion, this reference is not suitable with the purpose of the analysis made by the authors . Molecular type is an immunohistochemical classification or characterization of breast cancer for hormone receptor status or HER-2 gene overexpression and it correlates with the clinical outcome or the choice of systemic therapy (e.g. ESMO Educational sessions in Annals of Oncology vol 23 suppl 10, Sept 1,2012; National Comprehensive Cancer Network Guidelines version 4.2020; or Nature/NPJ Breast Cancer 6 March 2020). Molecular types also correlate with prognosis of breast cancer (2019 ASCO education books - Molecular testing in breast cancer by Jennifer K. Litton et al). Unfortunately, there is no relevant literature of molecular type of breast cancer in association with diagnosis delays. I am concern that this issue is the major issue of the manuscript.

Response: 

Thank you for your thoughts on the issue. We have decided to exclude molecular subtype variables out of the multivariate model and related references until there is further evidence to determine any potential influence it may have on diagnosis delays. As a result, we have also adjusted the results in Tables 3-5 and abstract and discussion sections for the exclusion of molecular subtypes in these analyses.

Minor issues

The authors edited the design as suggested.

Response:

Thank you for your favourable opinion.

---

## [Editor Report · Decision Letter 2]

26 Dec 2021

Delays in the presentation and diagnosis of women with breast cancer in Yogyakarta, Indonesia : a retrospective observational study

PONE-D-21-15358R2

Dear Dr. Hutajulu,

We’re pleased to inform you that your manuscript has been judged scientifically suitable for publication and will be formally accepted for publication once it meets all outstanding technical requirements.

Kind regards,

Evy Yunihastuti, MD

Academic Editor

PLOS ONE

Additional Editor Comments:

Please provide acceptable data set as recommended by Plos One, especially data that derived table 1-6. The dataset provided was only for figure 1, which is not acceptable. Please see http://journals.plos.org/plosone/s/data-availability#loc-recommended-repositories.

---

## [Editor Report · Acceptance letter]

5 Jan 2022

PONE-D-21-15358R2 

Delays in the presentation and diagnosis of women with breast cancer in Yogyakarta, Indonesia: a retrospective observational study

Dear Dr. Hutajulu:

I'm pleased to inform you that your manuscript has been deemed suitable for publication in PLOS ONE. Congratulations! Your manuscript is now with our production department. 

Kind regards, 

on behalf of

Dr. Evy Yunihastuti 

Academic Editor

PLOS ONE